# Limited progress in nutrient pollution in the U.S. caused by spatially persistent nutrient sources

Rebecca J. Frei[1,2]*, Gabriella M. Lawson[1], Adam J. Norris[1], Gabriel Cano[1], Maria Camila Vargas[1], Elizabeth Kujanpää[1], Austin Hopkins[1], Brian Brown[1], Robert Sabo[3], Janice Brahney[4], Benjamin W. Abbott[1]*

1 Department of Plant and Wildlife Sciences, Brigham Young University, Provo, Utah, United States of America, 2 Department of Renewable Resources, University of Alberta, Edmonton, Alberta, Canada, 3 United States Environmental Protection Agency, Washington, D. C., United States of America, 4 Department of Watershed Sciences and Ecology Center, Utah State University, Logan, Utah, United States of America

* rfrei@ualberta.ca (RJF); benabbott@byu.edu (BWA)

**Data Availability Statement:** All data files are available from the Environmental Protection Agency's National Aquatic Resource Surveys database (https://www.epa.gov/national-aquatic-

## Abstract

Human agriculture, wastewater, and use of fossil fuels have saturated ecosystems with nitrogen and phosphorus, threatening biodiversity and human water security at a global scale. Despite efforts to reduce nutrient pollution, carbon and nutrient concentrations have increased or remained high in many regions. Here, we applied a new ecohydrological framework to ~12,000 water samples collected by the U.S. Environmental Protection Agency from streams and lakes across the contiguous U.S. to identify spatial and temporal patterns in nutrient concentrations and leverage (an indicator of flux). For the contiguous U.S. and within ecoregions, we quantified trends for sites sampled repeatedly from 2000 to 2019, the persistence of spatial patterns over that period, and the patch size of nutrient sources and sinks. While we observed various temporal trends across ecoregions, the spatial patterns of nutrient and carbon concentrations in streams were persistent across and within ecoregions, potentially because of historical nutrient legacies, consistent nutrient sources, and inherent differences in nutrient removal capacity for various ecosystems. Watersheds showed strong critical source area dynamics in that 2–8% of the land area accounted for 75% of the estimated flux. Variability in nutrient contribution was greatest in catchments smaller than 250 km$^2$ for most parameters. An ensemble of four machine learning models confirmed previously observed relationships between nutrient concentrations and a combination of land use and land cover, demonstrating how human activity and inherent nutrient removal capacity interactively determine nutrient balance. These findings suggest that targeted nutrient interventions in a small portion of the landscape could substantially improve water quality at continental scales. We recommend a dual approach of first prioritizing the reduction of nutrient inputs in catchments that exert disproportionate influence on downstream water chemistry, and second, enhancing nutrient removal capacity by restoring hydrological connectivity both laterally and vertically in stream networks.

resource-surveys/data-national-aquatic-resource-surveys).

**Funding:** This work was supported by the National Science Foundation (awards EAR-2011439 and EAR-2012123 - BWA), and RJF is supported as a Vanier Scholar.

**Competing interests:** The authors have declared that no competing interests exist.

## Introduction

Nutrient pollution is one of the most serious and costly environmental crises [1–3]. The over-abundance of carbon (C), nitrogen (N), and phosphorus (P) in many terrestrial, aquatic, and marine environments has created eutrophic (over-nourished) conditions that harm human health and economy, degrade aquatic habitat, and exacerbate climate change [4–8]. Many human activities release nutrients into the environment, but agriculture, human sewage, industrial wastewater, and fossil fuel combustion are the primary sources [9–11]. Pioneering water quality legislation, such as the Clean Water Act in the U.S., has been effective at reducing discrete pollutant sources (e.g., wastewater and industrial effluent) in many regions, but nutrient legacies and continued diffuse nutrient sources from agriculture and atmospheric deposition sustain widespread eutrophication [12–18]. Additionally, two-way interactions between eutrophication and climate change are complicating mitigation efforts on both fronts [5,19–21]. Consequently, while major water quality improvements occurred in the U.S. immediately after the implementation of the Clean Water Act and other policies, recent efforts to reduce the occurrence of harmful algal blooms and nutrient loading more generally appear to have plateaued [22–24].

Continued improvement of water quality in the U.S. will require an understanding of both discrete and diffuse nutrient sources and sinks across the landscape. For example, legacy nutrients stored in soils and groundwater can create time lags between changes in nutrient inputs and the response of nutrient outputs [25–28]. These lags mean that current nutrient conditions can be a result of nutrient overuse from previous decades that arise as discrete nutrient sources from the past. There is compelling evidence that nutrient legacies are driving continued eutrophication in many environments, particularly those with long nutrient residence times in soil and groundwater [16,29–31]. However, it is unclear if spatiotemporal patterns of nutrient concentrations correspond with the legacy nutrient hypothesis (that nutrient pollution will only improve after decades of patient persistence)—or if contemporary management strategies aimed at reducing non-point source pollution could rapidly improve nutrient conditions [26,29,32–34]. The diversity of nutrient recovery trajectories following reductions in nutrient loading suggest that nutrient removal capacity, contemporary nutrient loading, and nutrient-specific transport dynamics are as or more important than nutrient legacy in determining nutrient fluxes in some watersheds [22,35–38].

While detailed inventories of nutrient loading are gaining accuracy and expanding in temporal and spatial coverage [39–42], simple diagnostic tools are still needed to assess nutrient trends and patterns in groundwater and surface water ecosystems [16,17,23,24,35,37]. Recently, Abbott and others [43] proposed a new analytical framework to assess spatiotemporal nutrient dynamics from infrequent, spatially extensive surveys (hereafter synoptic samplings). This synoptic framework measures the *persistence* of nutrient sources and sinks, the spatial scale or *patch size* of variation in nutrient concentrations, and the influence or *leverage* of individual subcatchments on watershed-level nutrient flux. The persistence of spatial patterns in the surface water network is quantified using rank correlations between instantaneous concentrations of nutrients at the same locations across samplings, with higher correlations indicating spatial patterns of nutrient sources and sinks are conserved over time [6,43–45]. The patch size of nutrient retention, removal, or release within an ecosystem can also be evaluated by analyzing the spatial scale (i.e., catchment size) where variance in concentration decreases rapidly or "collapses" toward the watershed mean [43,46,47]. Shogren and others [48] expanded on this framework by calculating the leverage of each subcatchment on overall watershed nutrient transport, allowing a first-order estimation of watershed-scale nutrient budgets and the identification of critical source areas from infrequent but spatially-extensive

synoptic samplings [49]. While these metrics cannot and do not seek to replace the important work of measuring nutrient inputs and outputs through time [22,50,51], they allow robust assessment of current nutrient state. When measured repeatedly through time, the persistence, patch size, and leverage metrics could enable detection of trends and shed light on the effectiveness of changes in management and the inherent nutrient removal capacity of different ecosystems [36,44,45,52].

In this context, we applied a synoptic ecohydrological approach [6,43,44,48] to data from infrequent but spatially extensive surveys covering the continental U.S. We focused on four primary questions: **1)** Is water quality improving across surveys (i.e., have C, N, and P concentrations decreased at national and ecoregion scales over the 2000–2019 period)? **2)** How effective are infrequent, spatially extensive surveys at characterizing water chemistry patterns and drivers in the U.S.? **3)** What is the net nutrient release and uptake throughout surface waters of the U.S. (i.e., where are the nutrient sources and sinks)? **4)** What landscape characteristics, including human land use, are associated with nutrient source and sink dynamics across the U.S.? We hypothesized that improved nutrient management, declines in atmospheric $NO_X$ deposition, and reductions in point source loads would cause decreases in nutrient concentrations through time—with some regional variation because of poorly constrained diffuse nutrient sources and variability in nutrient removal capacity and hydrological residence time [36,50,53–56]. Specifically, we hypothesized that these reductions in nutrient loading would create low spatial persistence among catchments (i.e., restructuring of nutrient concentration patterns) due to variable rates of nutrient decline and uneven elimination of point and diffuse nutrient sources. Conversely, spatial persistence could remain high despite overall decreases in concentration if local discrete and diffuse nutrient sources are less important than nutrient legacies, atmospheric deposition, and differences in nutrient removal capacity. Finally, we hypothesized that differences in land use and climate would result in distinct patch sizes among ecoregions [43], and that headwater streams and lakes would be disproportionately responsible for nutrient flux, in line with observations from other regions [43,48,57,58].

## Materials and methods

### Collection and analysis of water samples

We analyzed data collected as part of the National Aquatic Resource Surveys (NARS) by the U.S. Environmental Protection Agency (EPA) Office of Water to understand the hydrochemical patterns of water quality in the U.S. during seven surveys spanning 2000–2019 [59]. Detailed information about site selection and sampling can be found in the NARS documentation [59], and we include an abbreviated description below. All surveys used a weighted probability-based sample design for site selection that included 8,085 river and stream samples (hereafter referred to simply as streams) and 3,766 lake samples. The data were collected as part of the Wadeable Streams Assessment (WSA; 2000–2004), the National Rivers and Streams Assessment (NRSA; 2008–2009, 2013–2014, and 2018–2019), and the National Lake Assessments (NLA; 2007, 2012, and 2017). During the WSA (2000–2004), 1,392 samples were collected from 1,391 unique streams. From the NRSA, the 2008–2009 survey consisted of 2,320 samples collected from 2,114 streams, the 2013–2014 survey included 2,261 samples from 1,985 streams, and the 2018–2019 survey included 2,112 samples from 1,914 streams. The WSA shared 357 sites with the NRSA 2008 survey but had no shared sites with any of the other NRSA surveys. The NRSA 2008 survey shared 790 sites with NRSA 2013, and the NRSA 2013 survey shared 931 sites with the NRSA 2018 survey. For lakes, the NLA 2007 survey included 1,326 water chemistry samples from 1,148 lakes, the NLA 2012 survey included 1,230 samples from 1,127 lakes, and the NLA 2017 survey included 1,210 samples from 1,099 lakes. The NLA

2007 survey shared 406 sites with the NLA 2012 survey, and the NLA 2012 survey shared 495 sites with the NLA 2017 survey (See S1 and S2 Tables for summary statistics). Because catchment area data were not yet released by the EPA for the most recent surveys (NLA 2017 and NRSA 2018), we used catchment area for the sites that were sampled in any previous surveys. Using this method, we were able to concatenate area for 56% of the sites sampled in the most recent surveys for inclusion in the leverage and patch size analyses.

EPA site selection criteria changed for both lakes and streams between surveys, affecting the range of catchment sizes studied. In the 2007 NLA survey, only lakes larger than 4 hectares were sampled, while in the 2012 and 2017 surveys, lakes larger than 1 hectare (a minimum of 0.1 hectares of open water) and at least 1-meter deep were sampled. For streams, the 2004 WSA study included only small, wadeable streams that had an average Strahler stream order of 2. The 2008, 2013, and 2018 NRSA studies included a larger range of catchment sizes and sampled large rivers such as the Mississippi, Columbia, Colorado, and Snake Rivers. Because of these adjustments to site selection, the WSA and NRSA stream surveys had unequal variance for catchment size but the lake surveys were less affected (S1 Fig). We accounted for these changes in sampling strategies by using paired t-tests and rank correlations for the temporal analysis, which rely solely on repeat samples between surveys.

EPA field crews sampled lakes during the spring and summer, and streams were sampled in summer during baseflow conditions. At each site, a 4 L grab sample was collected from the surface at designated coordinates. Lake samples were collected from the euphotic zone (generally the first 2 meters or less as calculated by the Secchi Disk Transparency method) using an integrated sampler at the deepest point of natural lakes or from the midpoint of reservoirs. All samples were shipped on ice and analyzed within 24–48 hours of collection. Samples were analyzed for a suite of indicators including water chemistry, land use classification, physical habitat, benthic community, and fish ecology. In this study, we focused on dissolved organic carbon (DOC), nitrate ($NO_3^-$), total nitrogen (TN), and total phosphorus (TP), which are important water quality parameters relating to eutrophication, pollutant transport, aquatic habitat, and human drinking water [60–62]. While these parameters can serve as nutrient or energy sources [61,63,64], we refer to them as nutrients for convenience. We replaced zeroes in the concentration data with the measurement detection limit divided by 2.

### Temporal change in nutrients

We assessed temporal changes of nutrient concentrations (i.e., changes from survey to survey) at the continental and regional scales in the contiguous U.S. To account for variability in site selection across samplings, we used paired t-tests to evaluate changes between repeat sample sites only (S3 Table). This paired approach provided a sensitive and robust method to quantify change through time, which we visualized with boxplots (Fig 1). We log-transformed non-normal distributions (determined by histogram and normal Q-Q plot analysis) prior to running t-tests and used a decision criterion of $\alpha = 0.05$ to determine statistical significance. For sites that were sampled multiple times over individual survey periods (some locations were intentionally sampled repeatedly to quantify seasonal and inter-annual variability), we used the mean concentration of each parameter across the period.

We tested for temporal changes at the scale of the contiguous U.S. and for EPA Level I Ecoregions. We used ecoregions to group samples because these divisions are commonly used to implement ecosystem management strategies across federal agencies, state agencies, and nongovernmental organizations [65]. Additionally, ecoregions correspond with general ecosystem characteristics, including climate, vegetation, and land-use parameters [60], grouping areas with common ecological configurations and human disturbance patterns in North

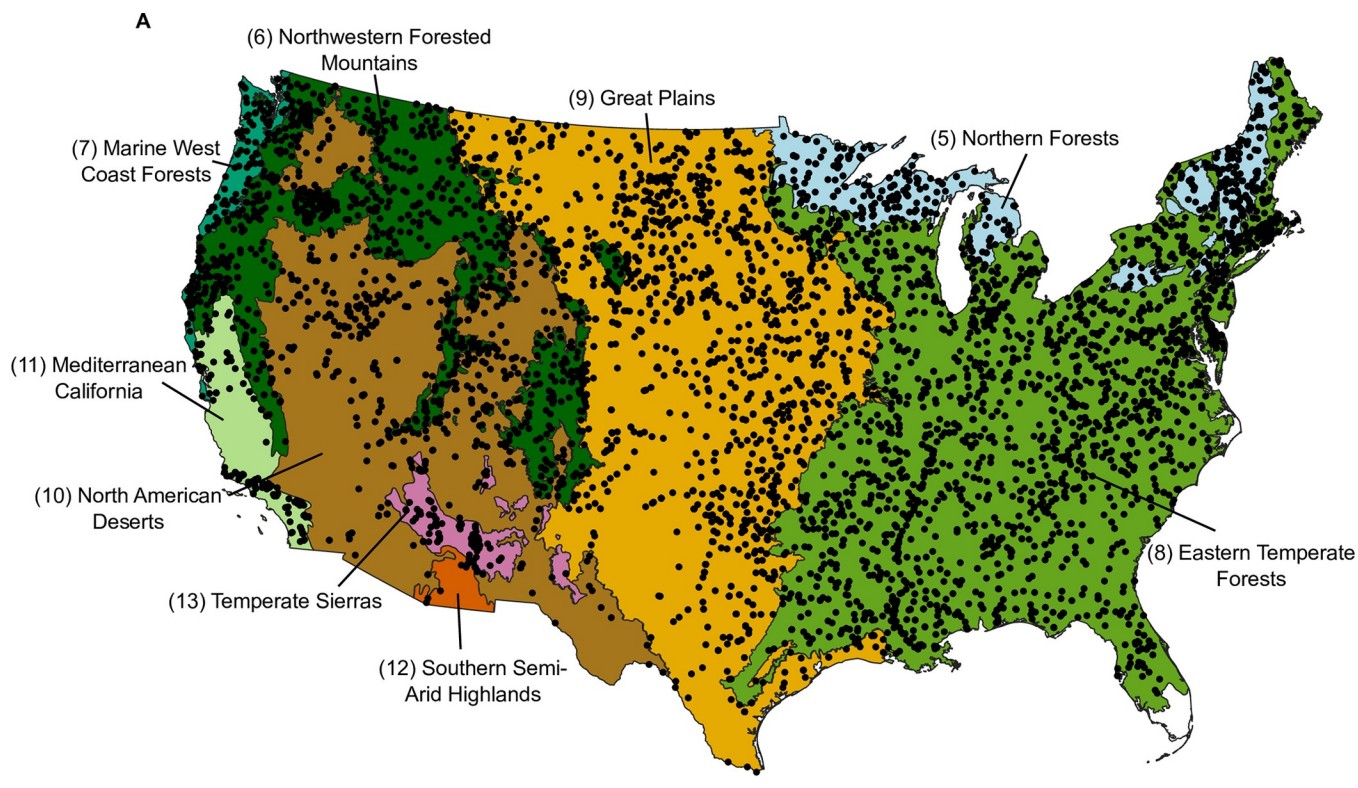

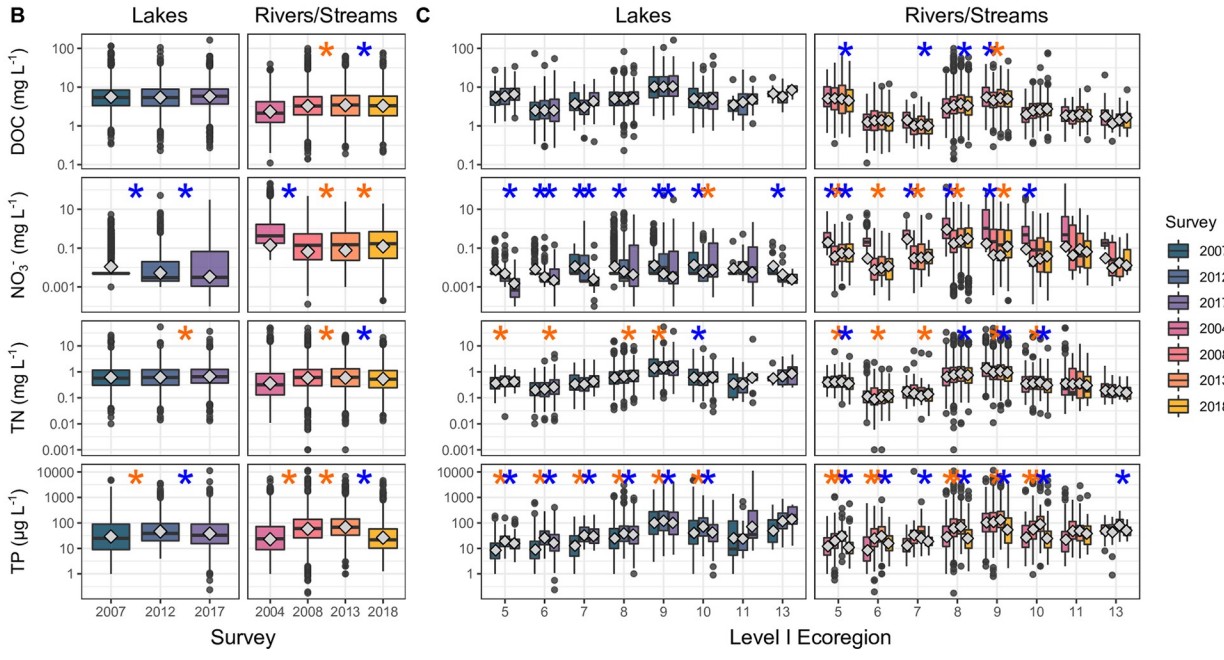

**Fig 1.** Map (panel A) of stream and lake sampling points and distributions of nutrient concentrations at continental (panel B) and ecoregion levels (panel C). The map (panel A) shows EPA NARS sample sites (black circles) included in the analysis. The map area is shaded by Level 1 Ecoregion, and those included in the analysis are labelled with the ecoregion name and number. Boxplots (panels B and C) show change in nutrient concentrations for all sites sampled in each survey. The middle horizontal line in the boxplots shows the median and the boxes represent the interquartile range. Silver diamonds represent mean concentration. Asterisks indicate significant differences between repeat samples as determined by paired t-tests (p < 0.05)—orange asterisks mark significant increases and blue asterisks mark significant decreases.

America. Dividing the analysis by ecoregion also reduced uncertainty from differences in specific discharge (the amount of runoff per unit area), which is one of the assumptions of the leverage analysis (see below). Our analysis focused on ecoregions 5–13, which are the main ecoregions found in the contiguous U.S., though ecoregion 12 did not have enough sampling points to calculate all the metrics. We divided the analysis by aquatic environment (i.e., lakes and streams) because of differences in nutrient dynamics between flowing systems with lower residence times (streams), and standing or lotic systems (lakes) with higher residence times and hence opportunities for uptake or release of different nutrients [61,66].

## Ecohydrological metrics

To understand how nutrients enter and propagate through freshwater landscapes, we calculated several ecohydrological metrics that evaluate spatiotemporal nutrient dynamics based on infrequent, spatially extensive surveys, such as the NARS [6,23,43,44,48]. These metrics quantify the spatial persistence, patch size, and leverage (relative influence) of nutrient sources and sinks, allowing comparison of nutrient state among different ecoregions and watershed sizes. When applied repeatedly, these metrics can indicate trends and effectiveness of management interventions. We present a general description and method of calculation for each of these metrics in the following paragraphs.

Spatial persistence is the degree to which spatial patterns within a watershed or region stay the same through time (e.g., the same locations always have higher nutrient concentrations relative to other locations). Spatial persistence indicates the representativeness of infrequent samplings, which can be associated with the synchronicity of changes in nutrient concentration through time, or the absolute difference in nutrient concentration among sampling locations [6,43,44,48]. For example, if nutrients have low spatial persistence (perhaps due to shifts in local sources or sinks) then analyses of patch size and nutrient contribution made for a single moment in time may not be representative of long-term nutrient status [6,43,67]. We calculated spatial persistence of water chemistry parameters for lakes and streams across the contiguous U.S. using Spearman correlations of nutrient concentration following Equation 3 from Abbott et al [43]. This nonparametric method compares only the rank of the data (e.g., the lowest value is assigned a rank of 1), making it robust to extreme values, which are common in aquatic chemistry. We calculated the persistence of nutrient concentrations at the national level and by ecoregion for each waterbody (i.e., lakes and streams) and nutrient. We use a spearman correlation of $\sqrt{0.5}$ (i.e., 0.71) as a qualitative threshold, indicating that more than 50% of the spatial pattern persisted across sampling dates for that nutrient (Fig 2A). To understand differences between lake and stream persistence, we regressed lake and stream ecoregion persistence with the null hypothesis that persistence values between waterbodies would not be different from each other (i.e., $R^2 \sim 1$, Fig 2B). We also analyzed the ions and physiochemical parameters (i.e., pH, turbidity, total suspended solids, and acid neutralizing capacity) included in the water chemistry dataset (S2 Fig).

Patch size analysis evaluates what spatial scales express the greatest variability in nutrient concentration, revealing the size of inherent ecosystem characteristics and human disturbances that create nutrient retention, removal, or release [36,43,46,47]. For example, if large-scale atmospheric deposition was the primary source of a nutrient, it would have relatively large patches with high nutrient concentration, whereas if discrete point sources were dominant, there would be relatively smaller patches of high and low nutrients. We identified the patch size of nutrient sources and sinks for lake and stream catchments by performing a variance collapse analysis on nutrient concentration data [43,48]. First, we scaled the nutrient concentrations by subtracting the ecoregion mean and dividing by the ecoregion standard

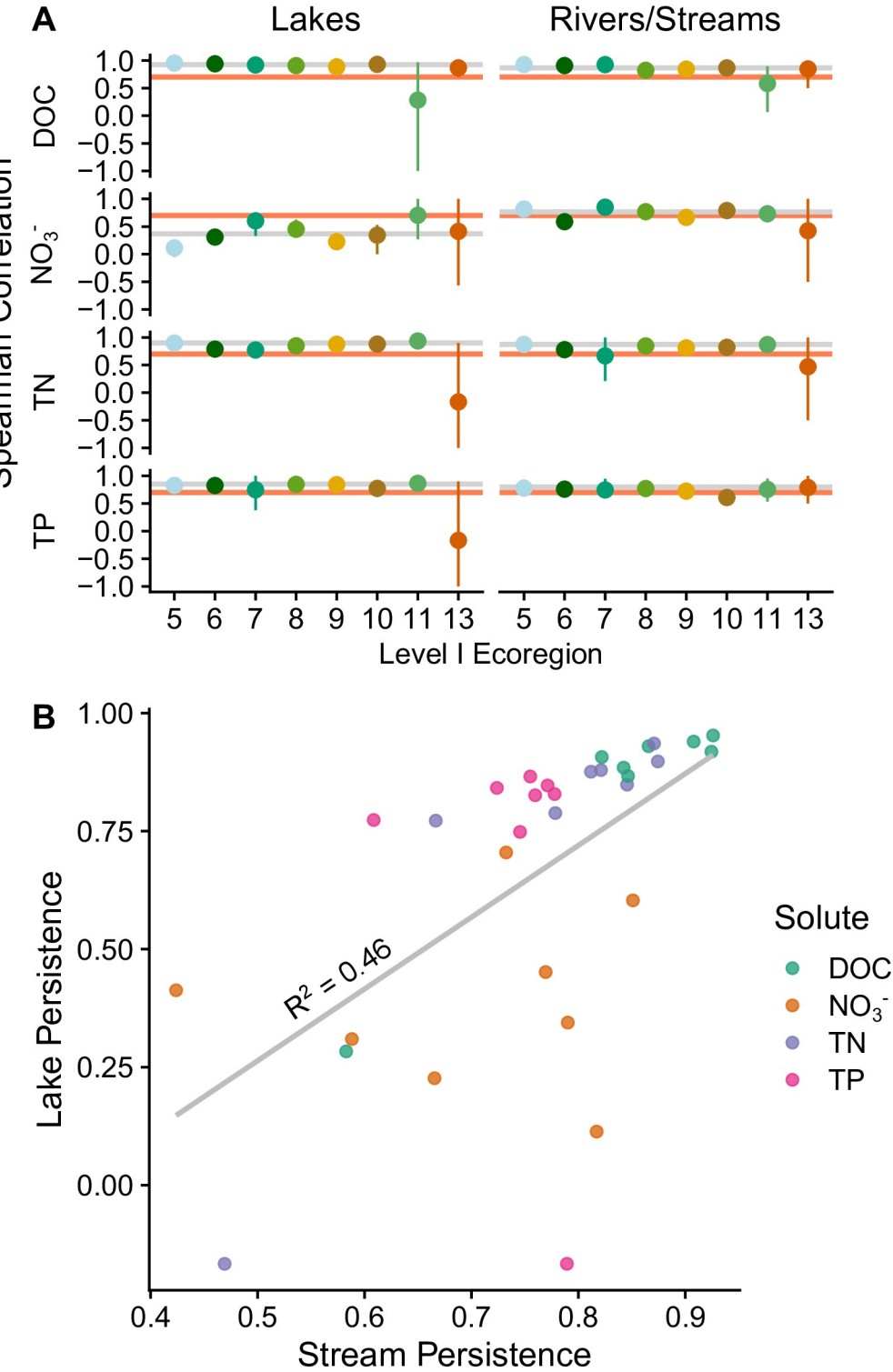

**Fig 2. Spatial persistence for C, N, and P by ecoregion for lakes and streams.** In panel A, points that fall above the orange line ($y = \sqrt{0.5}$) maintain more than half the spatial pattern for that parameter in pairwise comparisons of the two sampling dates. Horizontal gray lines show continental-scale persistence for that parameter. Panel B compares stream and lake persistence for each nutrient (i.e., each green dot represents mean DOC persistence for an ecoregion). The variable relationship between lake and stream persistence suggests that different factors affect persistence of nutrients in lakes and streams even within the same ecoregion.

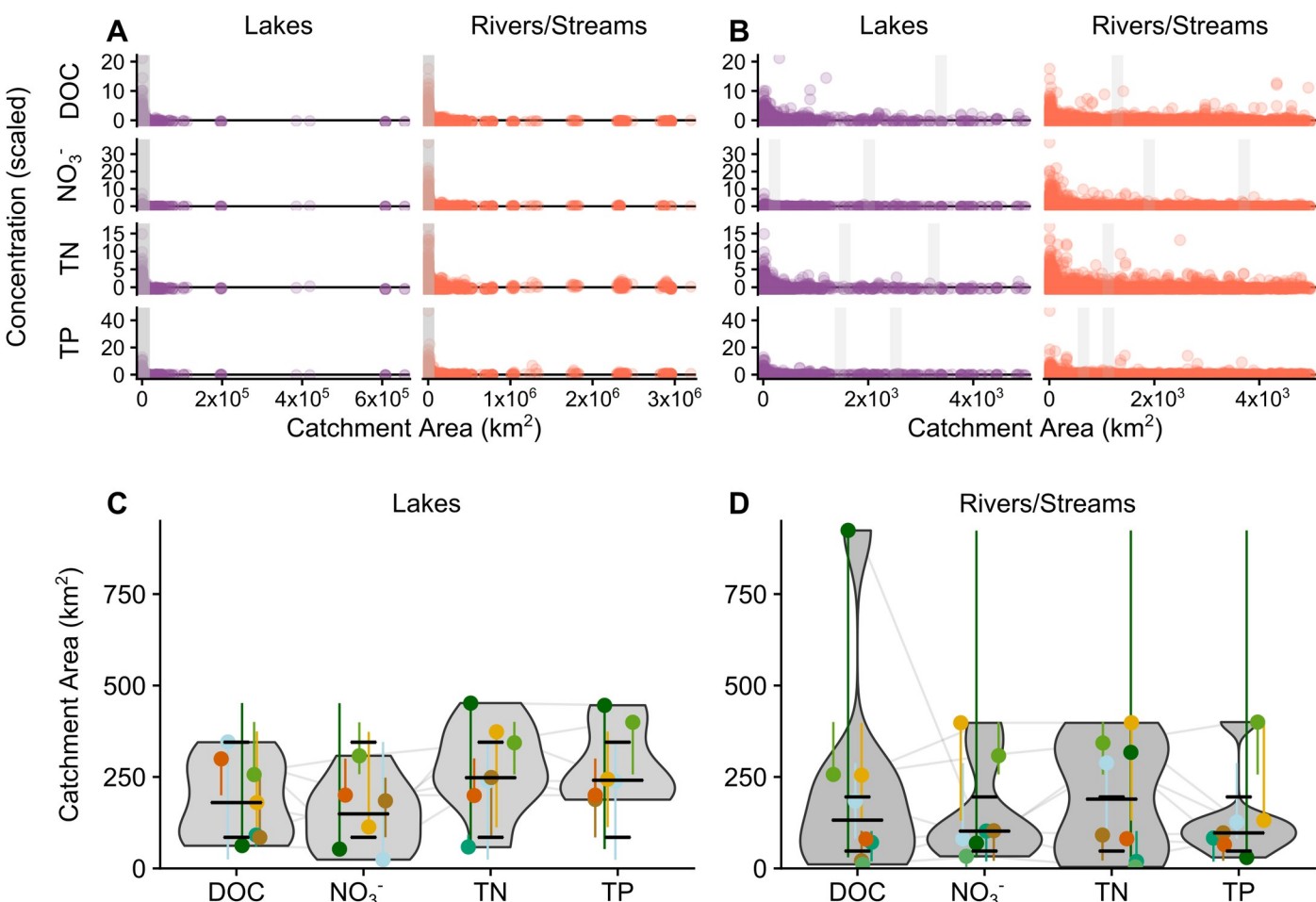

**Fig 3. Variance collapse for nutrients in lakes and streams in the U.S.** Panel A shows variance collapse thresholds (marked by vertical gray bars) of the entire national dataset. Panel B is a zoomed-in version of panel A showing variance collapse thresholds for C, N and P at the national scale. Violin plots show the distribution of ecoregion variance collapse thresholds for lakes (panel C) and streams (panel D) with short horizontal bars bounding the interquartile range and a longer horizontal bar showing the median. Colored point and whisker plots show the median and range of variance collapse thresholds for each ecoregion within the violin plots. Gray lines connect between ecoregion thresholds for different nutrients to show a general relationship among nutrients. Thresholds were determined by partial exact linear time (PELT) analysis.

deviation so that the concentration of each parameter had a mean of zero and a variance of 1. This allowed comparison of nutrients with very different mean concentrations. Next, we ordered the scaled concentration data by catchment size from smallest to largest and used the *changepoint* package in R to determine breakpoints in the data [68,69]. All breakpoints were then manually reviewed to check for false positives, which can occur when a single extreme point triggers the algorithm. The resulting breakpoints identified decreases of variance (i.e., collapse thresholds) with spatial scale, indicating the patch size of nutrient sources or sinks [43]. We assessed breakpoints for the entire national dataset (Fig 3A and 3B) and by ecoregion (Figs 3C, 3D and S3), except for ecoregion 12, which had few or limited samples.

Subcatchment leverage is a metric of the contribution of nutrients from each subcatchment relative to the overall watershed nutrient flux [43,48]. It combines concentration with catchment size (a proxy of discharge assuming specific discharge is similar across sites) to estimate whether each subcatchment increases (positive value) or decreases (negative value) the nutrient flux at the outlet, assuming conservative nutrient transport [43]. We calculated

subcatchment leverage by multiplying the difference in subcatchment and watershed outlet nutrient concentrations with the ratio of subcatchment area to watershed outlet area [43]. We used ecoregion instead of topographic watershed, defining the largest measured watershed within the ecoregion as the watershed outlet. This infringes an assumption in the calculation (all subcatchments are usually nested), but we chose this method for three reasons. First, even though each EPA sampling included thousands of locations, there were usually only a few points within individual watersheds, except for large rivers such as the Mississippi and Colorado. Second, many of the sampled locations were not nested within larger watersheds, such as those draining directly to the ocean or occurring within endorheic (terminal) basins. Small watersheds are influential in determining lateral nutrient transport, and are key indicators of local nutrient management and conditions [43,57,70]. Third, splitting by ecoregion allows grouping of sampling points with more similar land uses and climatic conditions than would be seen within a single large watershed. This allows for more sensitive comparison of nutrient source and sink dynamics in the terrestrial environment and aquatic network. For example, two sampling locations that are adjacent but occur in different watersheds may have more in common than nested watersheds that are more geographically separated. However, we acknowledge that some ecoregions are quite large (e.g., ecoregions 8 and 9) and encompass areas with diverse land use and climate. Ultimately, we opted to calculate subcatchment leverage primarily by ecoregion instead of large watersheds following the reasons above. However, we recognize that this is a compromise that limits some interpretation in our results. Consequently, we discuss primarily the relative differences among watersheds within an ecoregion.

Because discharge measurements were not available at most of the sampled locations, we assumed constant intra-ecoregion specific discharge [43,48]. We tested this assumption with discharge data for each ecoregion from USGS reference stations. We found comparable specific discharge across the sampling dates and for most ecoregions except ecoregions 6 and 7 (Northwestern Forested Mountains and Marine West Coast Forests), which showed substantial differences across sampling dates and between sites (S4 Fig). Raw leverage values were converted to percentages to estimate whether a specific subcatchment increased (leverage > 0) or decreased (leverage < 0) nutrient flux at the ecoregion outlet [48]. Thus, subcatchments with positive leverage indicated nutrient sources and negative leverage values indicated nutrient sinks. Very large leverage values can occur—for example, well over 100%—because some material is retained or removed as the water flows over and through the landscape before reaching the outlet. Conversely, estimates of leverage could be influenced by the differences in specific discharge mentioned above. For example, watersheds with high specific discharge tend to have shorter nutrient legacies [36]. Mean leverage values for a watershed (or in this case ecoregion) can estimate network-level mass balance. When the mean leverage is positive, this implies there has been nutrient removal within the surface water network (i.e., there are more sources than can be accounted for at the outlet), whereas a mean negative leverage value indicates production within the surface water network [48]. Given the multiple assumptions in the analysis, we interpreted the leverage values cautiously.

### Using machine learning to predict nutrient concentration and flux

Machine learning models are increasingly used to make hydrological predictions [71,72], and the most accurate versions tend to utilize ensemble models that combine inputs from independent algorithms before making final decisions [73–75]. Machine learning models can also be used to explore complex, non-linear relationships between predictor and target variables. In these situations, the behavior of the model is probed to identify the importance of predictor variables, such as catchment size or ecoregion, to predict various targets (in this case nutrient

concentration). While previous work has used single models in this kind of analysis, in this study we used four different machine learning models to produce more robust results and compare performance of the models. We included some of the most widely used models, including a decision tree regressor, a multi-layer perceptron, a random forest, and a gradient boosting regressor. For interpretation, we combined the feature importance and partial dependence plots of all four models into an ensemble, though we visualized the individual outputs to compare similarity qualitatively.

We used the ensemble model to explore relationships between the predictor variables (e.g., catchment characteristics and climate variables) and target variables (e.g., nutrient concentrations and subcatchment leverage, though we focus on concentrations because of the leverage assumptions mentioned above). Data for catchment characteristics were available through the NARS dataset including elevation; EPA ecoregion; and percent cover of agriculture, wetlands, forest, and urban land as estimated from areal NLCD land use classes. We planned to include land use data from each of the sampling campaigns to assess the legacy hypothesis directly (i.e., does land use from 5 or 10 years ago predict water quality better than current land use), but catchment characteristics were only available for the earlier surveys. Therefore, the models included concentration and leverage data from the first two lake and the first three stream surveys (i.e., NLA 2007 and 2012, and WSA 2004 and NRSA 2008 and 2013) as a case study. In addition to catchment characteristics, we extracted mean annual air temperature (MAAT) and precipitation from the WorldClim repository using the *raster* package in R. To account for differing biogeochemical processes among aquatic environments, we ran separate models for lakes and streams. We accounted for temporal differences between EPA surveys (e.g., whether the sample was collected during the 2007 or 2012 NLA) by including EPA survey as a categorical variable. Samples with missing stream chemistry data were removed from the dataset (<2% of samples with catchment characteristics data). Predictor variables were normalized to have values between 0 and 1. The model trained on a random subset of 80% of the concentration and leverage data and predicted the other 20%. Because the chemistry data were right-skewed, we binned the data and reweighted the probability with which each sample was considered, so that the model would more effectively learn to train on the feature data instead of predicting the most likely target values [76]. The multi-layer perceptron was the exception because the model could not use sample weights, thus the data were randomly drawn, agnostic to data distribution. We compared the predicted output versus the actual data and report metrics of model fit and relative importance of predictors.

Statistical and analytical methods were performed using R version 3.5.4 [77], except for the machine learning models that were completed using custom python (version 3.7) scripts and the scikit-learn package [78]. Code and data for the machine learning analysis are available at https://github.com/Populustremuloides/EPA_Synoptic_Water_Chemistry.git.

## Results

### Change in nutrient concentrations from 2000 to 2019

The temporal analysis of paired survey sites showed differing patterns among nutrients and between lakes and streams at the national and ecoregion scales (Fig 1). There were negligible changes in national and ecoregion lake DOC concentration ($p > 0.05$) during the 2007–2017 lake survey period ions (on average, ecoregions experienced a 7% increase in DOC concentration across surveys; Fig 1C). $NO_3^-$ steadily decreased in lakes nationwide from 2007–2017, with a 52% decrease from 2007 to 2012 and a 36% decrease from 2012 to 2017. Individual ecoregions followed a similar trend with an average decrease of 39% across ecoregions and surveys. Ecoregion 10 was the only outlier with a 36% significant increase in $NO_3^-$ concentration from

2012 to 2017. TN concentration did not change significantly nationwide from 2007 to 2012 but increased by 9% from 2012 to 2017 (p < 0.05). Individual ecoregions reflected this trend with slight but significant increases in ecoregions 5, 6, 8, and 9. TN concentration in ecoregion 10 followed a similar trajectory as $NO_3^-$: a decrease in concentration from 2007 to 2012 and then an increase from 2012 to 2017. TP significantly increased by 57% in lakes nationwide between 2007 and 2012 and then decreased by 17% from 2012 to 2017. Individual ecoregions exhibited similar patterns except ecoregions 11 and 13 which showed non-statistically significant increases in concentration from 2012 to 2017.

Streams followed the same pattern as lakes for TP, with stream TP concentration increasing significantly from 2000 to 2013 (156% increase from 2004 to 2008 and 17% increase from 2008 to 2013) and then decreasing by 61% from 2013 to 2018. All ecoregions showed significant decreases in TP concentration in streams from 2013 to 2018 except ecoregion 11, which did not have a significant change but still decreased by 22%. DOC and TN concentrations in streams also followed the pattern of increase from 2000 to 2013 and then decreased from 2013 to 2018, though the magnitude of change was less than for TP. For example, TP concentration increased from 2000 to 2013 by 62% across ecoregions on average, while DOC increased by 3% and TN decreased by 0.5%. Likewise, TP decreased by 46% on average across ecoregions from 2013 to 2018, but DOC and TN decreased by only by 3% and 9%, respectively. Additionally, there were fewer statistically significant changes across surveys and ecoregions for stream DOC and TN than for TP. Stream $NO_3^-$ concentration followed the opposite trend as stream DOC, TN, and TP, decreasing from 2004 to 2008 by 53% and then increasing by 16% and 61% in the subsequent surveys (i.e., 2008 to 2013 and 2013 to 2018). Summary statistics for all paired t-tests are reported in the supplementary material (S3 Table).

## Spatial persistence of nutrients

Most major nutrients (DOC, TN, and TP; Fig 2) and ions (S2 Fig) at the national scale had an average spearman rank correlation above the threshold of ρ = 0.71 (the square root of 0.5), representing the persistence of more than half of the spatial pattern. $NO_3^-$ was above the minimum threshold for streams but not for lakes (ρ = 0.77 and 0.37 respectively). Lakes and streams across ecoregions exhibited similar nutrient persistence for DOC, TN, and TP, but $NO_3^-$ was less persistent for lakes in most ecoregions (Fig 2). A simple linear regression between lake and stream ecoregion rank correlations showed a positive relationship with an $R^2$ of 0.46 (Fig 2B). This result is largely driven by low rank correlations for $NO_3^-$ and a few other low (e.g., ρ < 0.30) rank correlations for DOC, TN, and TP lake values from ecoregions 11 and 13. Spatial patterns of DOC concentration were persistent in every ecoregion except for ecoregions 11 and 13. Spatial patterns of $NO_3^-$ persisted for most streams except in ecoregion 6, 9, 11, and 13; and spatial patterns were not persistent for $NO_3^-$ concentration in lakes for all ecoregions. TP exhibited spatial persistence for all ecoregions except for lakes in ecoregion 7 and streams in ecoregion 10, 11, and 13.

## Patch size of nutrient sources and sinks

Scaled concentrations tended to converge towards 0 with increasing catchment area (Figs 3 and S3), indicating a representative sampling of source and sink tributaries across ecoregions. Variance collapse thresholds or patch size was nutrient dependent and ranged from 500 to 4,000 $km^2$ for national data and 4 to 924 $km^2$ for ecoregions. Median patch size of nutrient source and sink dynamics was less than 250 $km^2$ across nutrients and ecoregions, suggesting relatively fine-grained sources of variation even at continental scales (Fig 3). Lakes and streams had similar variance collapse thresholds for all major nutrients except TP, which had a much

smaller patch size for streams than for lakes. Likewise, ecoregion 9 showed almost a 3-fold difference in patch sizes for nitrogen and phosphorus (~125 km² vs ~350 km²). We note that not all ecoregions had sufficient sample size to determine variance collapse thresholds through PELT analysis (specifically ecoregions 12 and 13).

## Quantifying sources and sinks by ecoregion

Because the EPA NARS were not designed using a nested subcatchment framework, we used ecoregions to organize the leverage analysis, though this introduces some limitations as discussed in the methods. Subcatchment leverage generally followed the expected pattern wherein spatial variance of leverage decreased as subcatchment area increased (i.e., longitudinal averaging or mixing), and therefore most catchments had low leverage. Similarly, subcatchment leverage decreased with Strahler stream order. In other words, headwater catchments (Strahler stream order < 3; S5 Fig) largely controlled summer nutrient flux, including in downstream reaches (net change in longitudinal flux was largely ±1%, Figs 4, 5 and S6). Streams followed this spatial pattern the most closely, while lakes exhibited more spatial variability with some influential medium-sized subcatchments (Fig 4A). A notable exception to the headwater dominance occurred in the lower reaches of the Mississippi River where extremely large subcatchments (greater than 2,000,000 km²) exhibited surprising variability in subcatchment leverage for stream TN (Fig 4A). These high leverage values indicate additions of TN sufficient to change mainstem concentration. Most nutrients exhibited moderate to low leverage (both positive and negative), such that 92–98% of subcatchments accounted for less than 25% of ecoregion outflow concentration. Stated conversely, 2–8% of subcatchments accounted for 75% of the observed nutrient flux. An even smaller proportion of subcatchments (0.05–4%) had extremely large positive leverage values (> 100% leverage, and in some cases upwards of 3,000%), indicating critical source areas and inferring substantial retention or mixing of these nutrients in the surface water network (i.e., removal or dilution processes are needed to close the mass balance). The spatial analysis of the distribution of subcatchment leverage further highlighted that ecoregion nutrient dynamics were largely controlled by a handful of highly influential subcatchments (Fig 5).

At the continental scale, our results pointed to net removal or retention of N and P, and net conservative transport of DOC. Mean subcatchment leverage for $NO_3^-$, TN, and TP was positive and much greater than 0, indicating net removal or, at the minimum, retention in the surface water network (Fig 4B). Stated differently, the flux observed at the catchment—or in this case ecoregion—outflow was substantially lower than the inputs from the subcatchments, implying net removal. Mean leverage for DOC was close to zero (1% and 0.1% for lakes and streams), representing a neutral mass-balance (i.e., conservative mixing with little net production or removal).

## Modelling nutrients with catchment characteristics and climate

Nutrient concentrations showed a variety of relationships with catchment characteristics and climate based on the machine learning analyses (Fig 6). We synthesized the results from the four models—a decision tree regressor, multi-layer perceptron, random forest, and gradient boosting regressor—to democratize predictions of nutrient concentration. The average $R^2$ among the four models was 0.66 and 0.32 for lakes and streams respectively. Additionally, model predictions were compared to a random number generator that sampled from a normal distribution centered at the mean for each nutrient, and lake and stream models, on average, were 5.36 and 3.23 times better respectively at predicting water chemistry over the random number generator. Interestingly, lake models performed the best when predicting TP

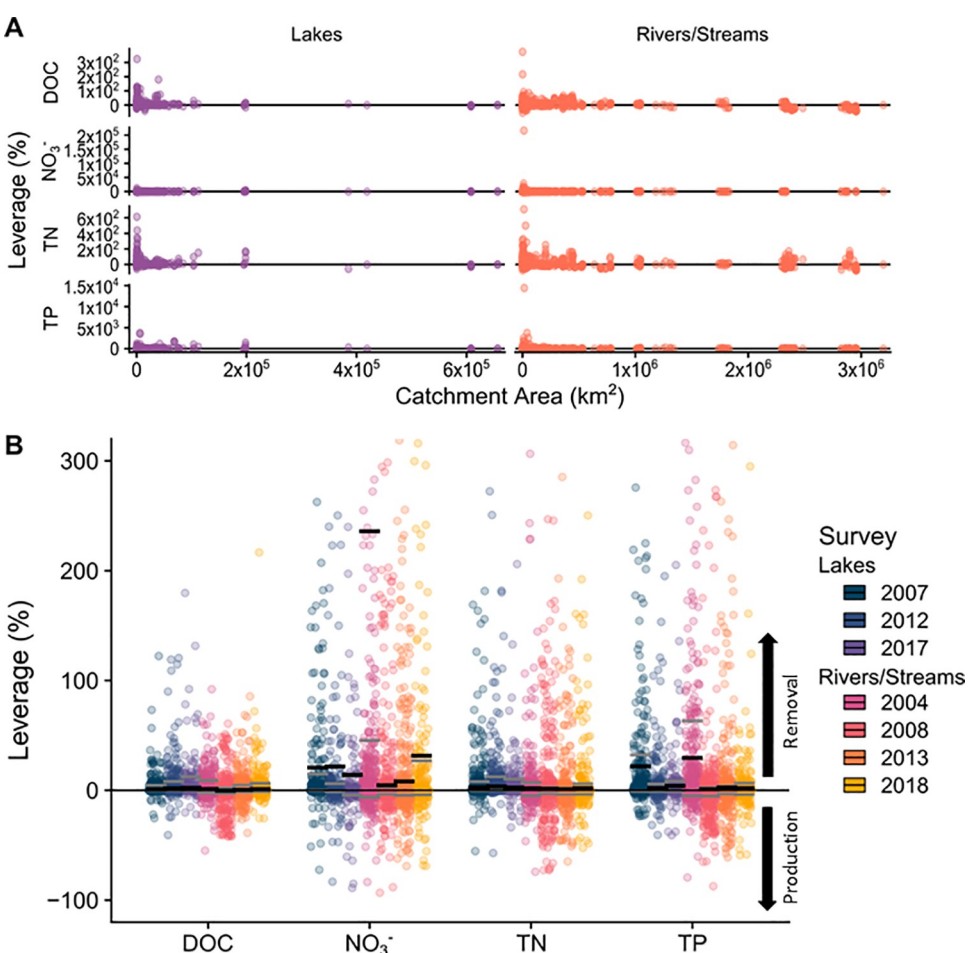

**Fig 4. Leverage for lakes and streams at the national level.** Positive leverage indicates nutrient sources and negative leverage indicates nutrient sinks (or weaker sources). In panel A, note the surprisingly low and high leverage values for DOC and TN in the large catchments of ecoregion 8. These leverage values correspond with reservoirs in the downstream reaches of the Mississippi River before entering the Gulf of Mexico. Panel B shows zoomed-in distributions of leverage values for nutrients in lake and stream catchments in the U.S. The horizontal gray bars represent the 95% confidence interval about the median, and the horizontal black bars represent the mean. A positive mean can be interpreted as the percentage of the total nutrient mass added to the stream network that did not leave the network (net removal), while a negative mean represents net production. Black arrows show that positive mean leverage values indicate nutrient removal and negative values indicate nutrient production.

concentration (mean $R^2$ = 0.81), but stream model performance was the worst for the same parameter (mean $R^2$ = 0.03). Lake models also generally outperformed stream models when predicting DOC, $NO_3^-$, and TN concentrations (S7A Fig).

One of the benefits of machine learning is the ability to interpret non-linear relationships between predictors and target variables. Partial dependency plots showed the relationship of major predictors for all four models, and agreement among plots helped strengthen or weaken our confidence in the observed relationships. For example, DOC concentration was strongly associated with precipitation in lake models and wetland cover in stream models (Fig 6A). DOC partial dependency plots (Fig 6B) showed that lake catchments with annual precipitation of ~400–600 mm had the highest DOC concentration, and that catchments with high annual precipitation (e.g., > 800 mm) were predicted to have the lowest DOC. Partial dependency plots for wetland cover in stream catchments showed a strong positive trend between wetland

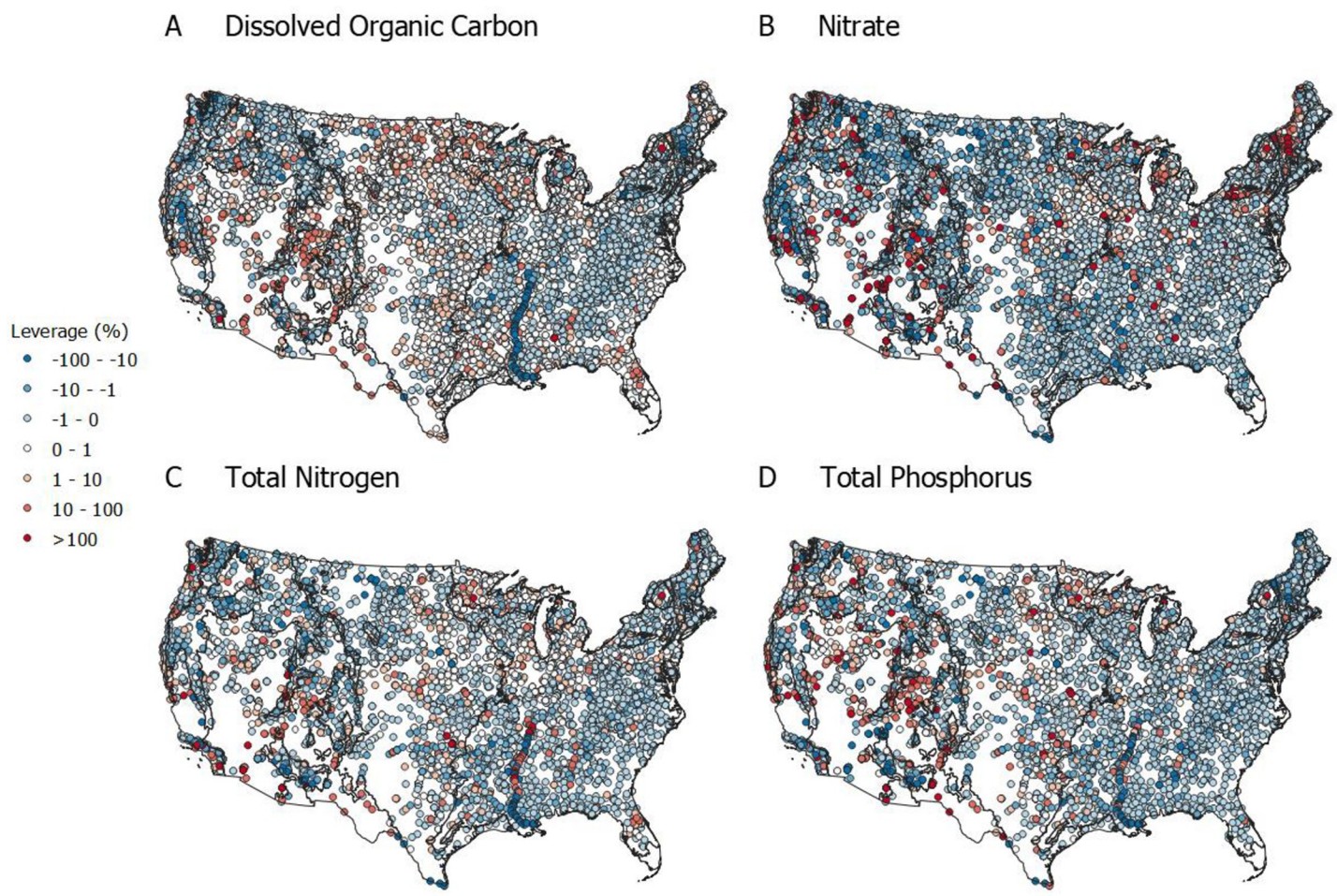

**Fig 5. Subcatchment leverage mapped across the contiguous U.S.** Points represent lakes and streams that were sampled over the study period (2000–2019). The points are colored by leverage value with cool colors representing negative leverage values (e.g., concentrations lower than the ecoregion outlet) and warm colors representing positive leverage values (e.g., concentrations higher than the ecoregion outlet).

cover and DOC concentration, though the trend is more logarithmic than linear. $NO_3^-$ lake models showed strong associations with agriculture, forest, and urban areas, and stream models were strongly associated with agriculture and elevation (Fig 6C). As to be expected, the models agreed that greater proportions of agricultural land cover increased $NO_3^-$ concentration in lakes and streams (Fig 6D). Partial dependency plots for elevation, however, showed less agreement among models. For streams, three of the models predicted highest concentrations at low elevations (e.g., 0–500 m), but the multi-layer perceptron model predicted an almost linear relationship with the highest concentrations at the highest elevations (> 2,000 m; Fig 6D). For lakes, elevation partial dependency plots also showed varying results, and the multi-layer perceptron model predicted a linear but opposite relationship to the stream model (i.e., lowest concentrations at highest elevations). TN was also associated with catchment characteristics such as forests, agriculture, and elevation, though none of these were particularly strong predictors on their own (Fig 6E). TP concentration in lakes was strongly associated with annual precipitation and to a lesser extent with elevation. Streams were also loosely associated with precipitation and MAAT (Fig 6F). Precipitation had a generally negative relationship with TP concentration, and like DOC, the highest concentration occurred in moderately

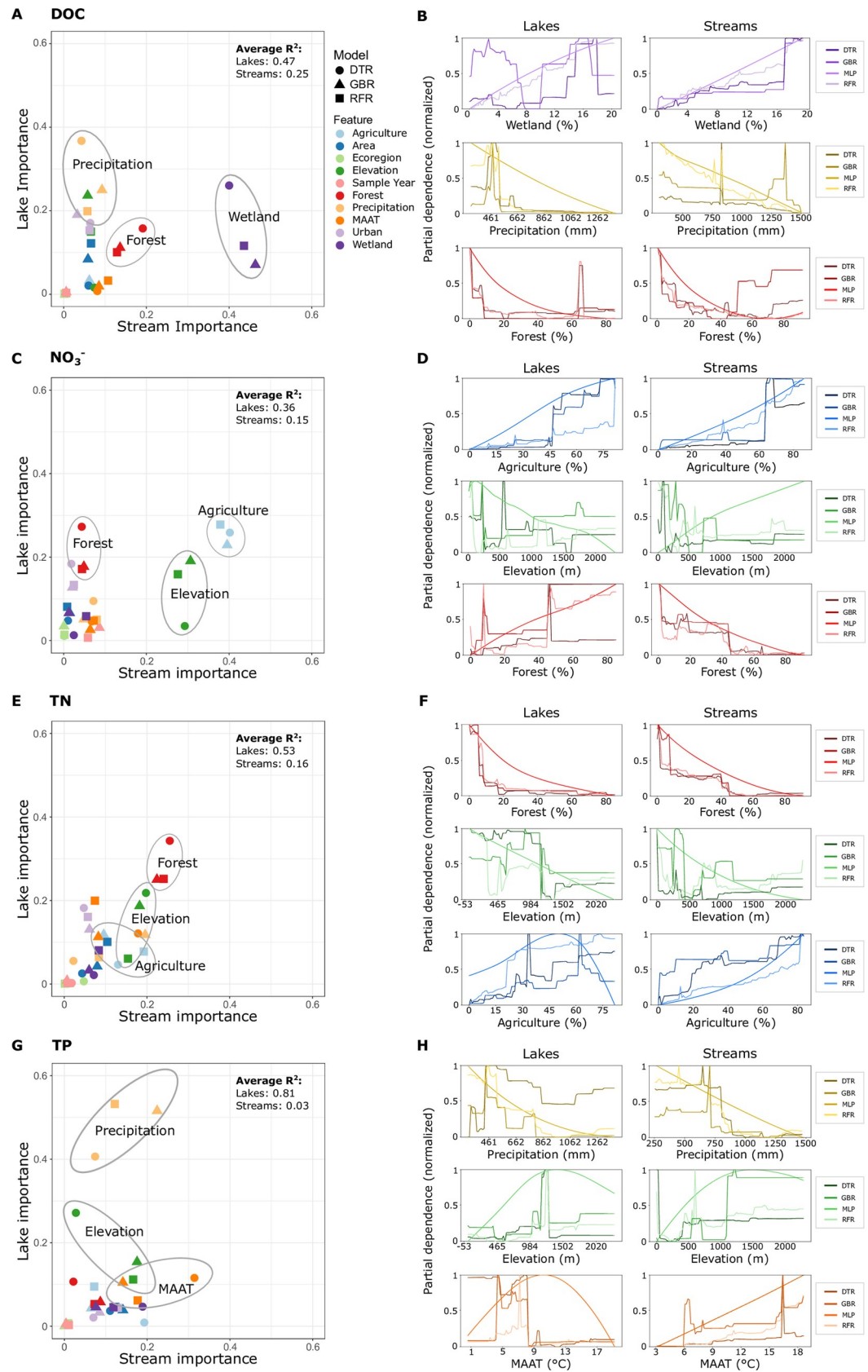

**Fig 6. Machine learning models illustrate the relationship between catchment characteristics, climate variables, and nutrient concentration.** Biplots between lake and stream model predictor importance (panels A, C, E, and G) show the most important variables that predict nutrient concentration. Partial dependence plots (panels B, D, F, and H) show the relationship between the dominant model predictors (e.g., mean annual air temperature (MAAT), agriculture, etc.), and nutrient concentration. The y-axes of the partial dependence plots have been normalized between 0 and 1 for model comparison. Four different machine learning models are shown to democratize the results and interpretation.

wet climates (e.g., 400–800 mm; Fig 6H). We also modeled subcatchment leverage to gain insight into the drivers of flux dynamics and observed similar trends. The subcatchment leverage model results are reported in the supporting information (S7B, S8 and S9 Figs).

## Discussion

In this study, we used spatially extensive stream and lake chemistry data collected by the EPA to test hypotheses about trends and drivers of surface water nutrient concentrations in the U.S. Contrary to our hypotheses, we did not observe monotonic declines in most nutrient concentrations despite known decreases in nutrient inputs for many regions [23,40]. In the following sections, we draw on various results from this study and the broader literature to evaluate what these complex spatiotemporal patterns could mean for nutrient management and water quality in the U.S. and elsewhere.

### Have excess nutrients peaked or plateaued?

Environmental legislation such as the U.S. Clean Water Act has been largely successful in reducing point source pollutants, including wastewater effluent, feedlot outflows, and industrial discharge [12]. These decreases in nutrient inputs are having measurable effects on some water quality parameters, which should be recognized and celebrated [24,79,80]. However, our study and other recent work [14,23,81] highlight that the goal of the Clean Water Act "to restore the chemical, physical, and biological integrity of U.S. waters" has not been fully realized. From a management perspective, one of the primary questions is now whether the lack of progress in nutrient concentrations is a temporary condition preceding a gradual decline or a sign of stabilization at a new plateau.

Though reductions in $NO_x$ emissions and wastewater nutrient release have improved nutrient pollution in some areas, persistent loading from diffuse sources and long-term nutrient legacies have hindered progress [35,54,56,82]. Indeed, the results in this study showed that nutrient concentrations in lakes and streams nationwide either increased or remained static until the most recent NLA 2017 and NRSA 2018 surveys, which exhibited downward trends for most nutrients. This downward trajectory is certainly encouraging, but it largely represents an incremental step toward the range of the earliest surveys' nutrient conditions, and could be due to interannual hydrological variability or non-stationarity more generally associated with other factors. One explanation for this water quality plateau could be that we will not see major improvements until nutrient legacies are depleted [16,80]. If legacies are the cause of the lack of progress, current nutrient management efforts should be sustained where in place and applied to areas that currently are not managed.

Another explanation for the nutrient plateau is that nutrient inputs have not decreased sufficiently to tip watersheds into net nutrient decline [26,35,83]. If this is the case, we need a major change to our overall management approach regarding method, intensity, and/or location. For example, this study and much previous work suggests focusing best practices in the watersheds acting as critical source areas [58,84]. While this may not seem tractable at regional to continental scales, we suggest that it would be much more cost effective to invest in adequate

monitoring and spatiotemporal analysis prior to targeted intervention in the subcatchments with highest leverage.

In either scenario (i.e., peak or plateau), the spatiotemporal complexity of water quality trends requires renewed innovation in data analysis to assess the effectiveness of current conservation measures. To determine whether the current lack of progress is a transition into a new water quality era or not, we need creative analysis of monitoring data and new modeling approaches to assess mass-balance [38,79,85]. New methods for combining spatiotemporal water chemistry monitoring with nutrient inventories and reactive transport models [17,34,37,44,86] could test many of the hypotheses raised by this study. Likewise, repeated and spatially extensive sampling such as these EPA surveys combined with remote sensing and nutrient retention modeling could be sensitive indicators of progress and management priorities [18,34,43,60,87]. Because the question of whether current management is adequate to result in ecological recovery is so fundamental to the goals of the Clean Water Act, we suggest more specific measures later in the discussion to hopefully generate more rapid water quality improvements.

## Spatially persistent nutrients across ecoregions

We found that nutrients in many ecoregions are spatially persistent, increasing confidence that our analysis is representative of spatial nutrient dynamics and are not just an artifact of climatic variability. This high degree of spatial persistence could be evidence of three non-exclusive factors. First, it could be due to nutrient legacies. Even though nutrient inputs have changed through time in certain regions of the country, the long residence time of water and nutrients in watersheds could result in persistent spatial patterns [27,88,89]. Second, ecological differences in watershed characteristics could create large differences in nutrient retention, contributing to spatial persistence. There is increasing evidence that these watershed-scale differences in retention are associated with subsurface characteristics, including the soil, vadose zone, and groundwater [36,37]. Third, the location of nutrient sources may not have changed as much as expected. The locations that are best suited for agricultural or urban development are not randomly distributed in space, nor can they change easily through time [90]. Though practices have improved through time and atmospheric deposition has decreased in some areas [90], the relative distribution of nutrient loading and nutrient retention appears to have persisted.

These results are consistent with national inventories that found that areas of high N inputs and surpluses were the same from 2002 to 2012 [50]. High spatial persistence further increases our confidence in the efficacy of infrequent, spatially extensive surveys [6]. While large surveys, such as the EPA NARS dataset require substantial investment of time and analytical capacity, the spatial persistence of most nutrients demonstrates the efficacy of the NARS design and provides critical information to improve watershed management. Citizen science approaches could allow continued and even expanded synoptic sampling at low cost, while also providing valuable experiences that can improve public understanding and willingness to support water conservation and protection measures [35,45,91–93].

One important exception to the pattern of spatial persistence was $NO_3^-$ in lakes. This result was particularly interesting because the opposite was observed for $NO_3^-$ in streams and for TN in both waterbody types. We expect that low spatial persistence of $NO_3^-$ is primarily due to high temporal variability of biogeochemical and hydrological processes in lakes (e.g. stratification and turnover) [48,94]. Indeed, *in situ* nitrification and denitrification rates in lakes are highly variable and depend on water depth, residence time, and microbial community [94,95]. Moreover, lakes were sampled across the spring and summer months when lake stratification

and turnover could have contributed to low persistence of $NO_3^-$, which is particularly sensitive to redox conditions [36,96]. These dynamics could mean that more frequent spatially extensive sampling of lakes is needed to adequately represent short- and long-term variability, and consequently there is inherent uncertainty in lake nutrient trends identified here.

## Locally sourced nutrients

Our analysis of the spatial scale of variance found that landscape patches that contribute or retain nutrients were relatively small (~250 km$^2$), though this varied across ecoregions (Figs 3 and S6). We note that patch size can also change on event, seasonal, and inter-annual time-scales [43,48,97–99], and we do not imply that the patch sizes determined here are inherent or permanent. However, the consistent pattern of patches at the tens to hundreds of square kilometer scales reinforces the conclusion that these intermediate scales are ecologically relevant to the release and retention of nutrients in watersheds of the U.S. From an administrative perspective, these scales (~250 km$^2$) correspond to regional governmental units such as municipalities, counties, or states, emphasizing the importance of collaboration with local stakeholders to develop and implement durable and effective management approaches. Differences in patch size among nutrients offered some insight into possible drivers of specific nutrients. In particular, patch size of TP sources and sinks were consistently smaller than TN, highlighting potential hotspots of legacy P eroding from the landscape (e.g., in areas surrounding concentrated animal feeding operations or P-rich minerals).

## Prioritizing highly influential catchments to improve water quality

This study provides further evidence that U.S. headwater catchments set the initial nutrient state as the majority of downstream reaches were neither major sources nor sinks in the stream network [42,100]. A surprising exception to this spatial pattern was in the lower Mississippi River Basin where nutrient concentrations suggested a strong removal signature along the mainstem of the river. Some of the major control points occurred at the confluence of the Arkansas and Ohio Rivers with the Mississippi River, as well as further downstream where retention reservoirs have been constructed for nutrient capture. The continental scale analysis of nutrient source and sink dynamics raised a long-standing question for water quality management: how much of the observed changes in nutrient concentrations can be accounted for by decreased retention versus increased loading in a catchment? In other words, would more water quality improvements be possible if we decreased nutrient loading instead of (or in conjunction with) focusing on landscape management strategies such as constructing riparian zone buffers? Our analysis of subcatchment leverage showed that most catchments exhibit neutral effects on overall nutrient export, as such downstream nutrient loads into surface water largely maintain observed nutrient concentrations, even in many areas where nutrient inputs are known to be quite large (Figs 4 and 5). This suggests that most catchments are highly efficient nutrient processors and could respond favorably to decreased loading. To reduce waterbody impairment, our data support a dual approach that first and foremost prioritizes reducing nutrient inputs to catchments that exert disproportionate influence on downstream water chemistry and increasing hydrological connectivity within stream networks (where effectual) to enhance nutrient removal capacity [36,101].

Because headwaters largely determine downstream water quality [102], regional efforts to decrease nutrient inputs and surpluses could yield promising results [42,49,103]. Indeed, localized management efforts in the Chesapeake Bay clean up and elsewhere in the U.S. have led to improvements in water quality that can be replicated elsewhere [50,56,104–107]. This emphasizes the need to craft regional solutions and ensure entities or individuals at the local level are

empowered to achieve them (i.e., subsidiarity). The needed information and resources to achieve further improvements include: **1)** continued innovation in nutrient use efficiency in agriculture [55], **2)** educational outreach and incentives to further improve nutrient management on farms [108,109], **3)** documenting nutrient inputs and surpluses across the landscape to identify inefficiencies in the handling and use of N and P [50], **4)** identifying areas of the landscape disproportionately degrading downstream water quality [80,98], and **5)** establishing clear watershed-wide nutrient goals towards which local communities can strive [81,110]. Certainly, to assess the effectiveness of local decisions in achieving water quality goals, standardized tracking of management actions (e.g., fertilizer use, cover crops) and continued water quality monitoring is essential. The latter is generally fulfilled by the EPA National Aquatic Resource Surveys (complimented by a network of intensively monitored sites from other state and federal organizations), and the former is largely addressed with recent nutrient inventory work though standardized accounting of other management actions like implementation of cover crops and tillage practices across the landscape [50,53].

## Conclusions

This study advances the understanding of spatial and temporal nutrient dynamics across the U.S. and encourages water quality managers to consider the watershed context and implement nutrient reduction strategies locally. Our analysis of the EPA National Aquatic Resource Surveys at the national and ecoregion levels revealed spatially persistent nutrient dynamics in most regions of the country. We also determined the patch size of nutrient sources and sinks and identified likely areas of water quality degradation across the United States. These analyses indicated a need for local interventions within highly influential catchments, in addition to general reductions of nutrient loading. Most catchments are already efficient nutrient processors; although nutrient retention pathways are not linear functions, we expect that reducing nutrient loads will lead to a net reduction in nutrient fluxes and concentrations at regional and national levels.

## Supporting information

**S1 Fig. Variability of catchment sizes among the National Aquatic Resource Surveys.** Boxplots show the interquartile range and median (middle horizontal line) and notches show the 95% confidence interval around the median.
(DOCX)

**S2 Fig. Spatial persistence at the national level for all major nutrients and ions.** The red line ($y = \sqrt{0.5}$) delineates where at least half of the spatial pattern is preserved across samplings.
(DOCX)

**S3 Fig. Variance collapse thresholds determined by PELT analysis for each ecoregion.**
(DOCX)

**S4 Fig. Variability of specific discharge at neighboring USGS reference stations for the EPA National Aquatic Resource Surveys by ecoregion.** Boxplots show the interquartile range and median (middle horizontal line) of discharge data. Notches show the 95% confidence interval around the median.
(DOCX)

**S5 Fig. Boxplots of Strahler stream order and catchment area.** The horizontal gray line marks where the 250 km$^2$ patch size lies in relation to stream order. Stream orders $\leq 3$ are

considered headwater streams.
(DOCX)

**S6 Fig. Subcatchment leverage by ecoregion.**
(DOCX)

**S7 Fig. R-squared values for machine learning models.** Models predicted nutrient concentration (plot A) and subcatchment leverage (an estimate of nutrient flux; plot B) using catchment characteristics and climate variables from the EPA National Aquatic Resource Surveys and WorldClim database.
(DOCX)

**S8 Fig. Machine learning models illustrating the relationship between catchment characteristics, climate variables, and subcatchment leverage (an estimate of nutrient flux).** Biplots between lake and stream model predictor importance show the most important predictors among the models predicting subcatchment leverage.
(DOCX)

**S9 Fig. Partial dependence plots show the modeled relationship between subcatchment leverage (an estimate of nutrient flux) and the primary predictors for lakes and streams.** They show how the predictions partially depend on the input variables of interest and are able to depict whether the relationship is linear, curvilinear, or a step function. The plots above show the partial dependence plots for the major predictors of DOC (A), $NO_3^-$ (B), TN (C), and TP (D) subcatchment leverage for lakes and streams.
(DOCX)

**S1 Table. Summary of the number of sites in each EPA National Aquatic Resource Survey (NARS).**
(DOCX)

**S2 Table. Nutrient summary statistics per ecoregion.**
(DOCX)

**S3 Table. Paired t-test results (p-values) and percent change from repeat sample sites between surveys.**
(DOCX)

## Acknowledgments

Data were collected by the U.S. Environmental Protection Agency, Office of Water. We thank Sarah Stackpoole and Jiajia Lin for internal review of the manuscript. The views presented here are those of the authors and do not represent official views or policy of the U.S. Environmental Protection Agency or any other U.S. federal agency. Any use of trade, firm, or product names is for descriptive purposes only and does not imply endorsement by the U.S. Government.

## Author Contributions

**Conceptualization:** Rebecca J. Frei, Gabriella M. Lawson, Adam J. Norris, Maria Camila Vargas, Elizabeth Kujanpää, Austin Hopkins, Robert Sabo, Benjamin W. Abbott.

**Data curation:** Rebecca J. Frei, Gabriella M. Lawson, Adam J. Norris, Maria Camila Vargas, Elizabeth Kujanpää, Austin Hopkins, Robert Sabo, Benjamin W. Abbott.

**Formal analysis:** Rebecca J. Frei, Gabriella M. Lawson, Adam J. Norris, Gabriel Cano, Brian Brown, Benjamin W. Abbott.

**Investigation:** Rebecca J. Frei, Gabriella M. Lawson, Adam J. Norris, Robert Sabo, Benjamin W. Abbott.

**Methodology:** Rebecca J. Frei, Gabriella M. Lawson, Adam J. Norris, Gabriel Cano, Elizabeth Kujanpää, Austin Hopkins, Brian Brown, Robert Sabo, Janice Brahney, Benjamin W. Abbott.

**Project administration:** Robert Sabo, Benjamin W. Abbott.

**Resources:** Robert Sabo, Benjamin W. Abbott.

**Supervision:** Robert Sabo, Janice Brahney, Benjamin W. Abbott.

**Visualization:** Rebecca J. Frei, Adam J. Norris, Maria Camila Vargas, Austin Hopkins, Brian Brown, Janice Brahney, Benjamin W. Abbott.

**Writing – original draft:** Rebecca J. Frei, Gabriella M. Lawson, Adam J. Norris, Gabriel Cano, Maria Camila Vargas, Elizabeth Kujanpää, Austin Hopkins, Brian Brown, Robert Sabo, Janice Brahney, Benjamin W. Abbott.

**Writing – review & editing:** Rebecca J. Frei, Gabriella M. Lawson, Adam J. Norris, Gabriel Cano, Maria Camila Vargas, Elizabeth Kujanpää, Austin Hopkins, Brian Brown, Robert Sabo, Janice Brahney, Benjamin W. Abbott.

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
