## [Decision Letter · Decision Letter 0]

12 May 2021

PONE-D-21-01607

Limited progress in nutrient pollution in the U.S. caused by persistent nutrient sources

PLOS ONE

Dear Dr. Frei,

Thank you for submitting your manuscript to PLOS ONE. First, I apologize for the delay in returning this manuscript due to the difficulty in getting reviewers' reports. Several reviewers agreed to review the manuscript but never provided comments; thus, I had to seek more reviewers, causing further delay. Second, after careful consideration, we feel that it has merit but does not fully meet PLOS ONE’s publication criteria as it currently stands. Therefore, we invite you to submit a revised version of the manuscript that addresses the points raised during the review process.

Here are some additional comments:

One reviewer has made excellent comments, including (1) using watersheds than ecoregions as the latter does not always follow watershed boundaries and (2) making the study novel by using the reviewer's suggestion on restructuring the manuscript.

The dataset used in the study is till 2012. Is there no data available after 2012?

The words persistence and leverage are not common in biogeochemistry research. I noticed that you indicated a brief definition in the introduction. If possible, attempt to simplify the definition and include it in the results and discussion sections, as appropriate.

I found the article difficult to read (including methods), see what could be done to facilitate reading.

One reviewer included a Word file with comments.

We look forward to receiving your revised manuscript.

Kind regards,

Gurpal S. Toor, Ph.D.

Academic Editor

PLOS ONE

2. We note that Figures 1 and 5 in your submission contain map/satellite images which may be copyrighted.

a. You may seek permission from the original copyright holder of Figures 1 and 5 to publish the content specifically under the CC BY 4.0 license. 

Reviewers' comments:

Reviewer's Responses to Questions

**Comments to the Author**

1. Is the manuscript technically sound, and do the data support the conclusions?

Reviewer #1: Yes

Reviewer #2: Yes

Reviewer #3: Yes

2. Has the statistical analysis been performed appropriately and rigorously? 

Reviewer #1: Yes

Reviewer #2: Yes

Reviewer #3: Yes

3. Have the authors made all data underlying the findings in their manuscript fully available?

Reviewer #1: Yes

Reviewer #2: Yes

Reviewer #3: Yes

4. Is the manuscript presented in an intelligible fashion and written in standard English?

Reviewer #1: Yes

Reviewer #2: Yes

Reviewer #3: Yes

5. Review Comments to the Author

Reviewer #1: I was really excited to read this paper about the persistence of nutrient loading across the United States. The paper introduction raises the interesting question of variation in nutrient persistence pathways: how surface water responds to reductions in nutrient inputs might follow various pathways depending on X, Y, Z. Unfortunately, the authors didn’t address this exciting question with their analyses – instead the analysis focused on understanding what land use was associated with concentrations of N, P and DOC. I think there is a lot of potential here, but I struggled with the disconnect between the gap articulated in the introduction and the analyses. I think there are two pathways forward: 1) keep the analysis as is. The gap could then be that there have been analyses of controls on nutrients, but they could be affected by temporal variation among sampling dates. This analysis tests that assumption – for the most part spatial patterns are persistent, so we can really trust the machine learning analysis. I think this would be a less interesting paper. We already know that land use affects nutrient concentrations, that these affects differ among nutrients. I wasn’t really clear what was novel about that part of the analysis. 2) What I think would be a more novel contribution would be to explore different nutrient trajectories, and where those are happening. I could imagine a similar ML approach, but using changes in nutrient concentrations (or changes in rank within a watershed?) as the response variable. Would allow you to address – what are the different pathways? What land uses are associated with increases/decreases? Are there differences among watersheds (or regions, but see my comments on that below) that are related to regional/local nutrient management approaches? For example, it would be interesting to see if agriculture is associated with increases in some regions and decreases in others (or no change in others) and if that were linked to differences in policies, etc. I recognize that this would be a substantially different paper, but I think the questions raised in the introduction are exciting and addressable using this dataset with some modified analyses.

Some more major comments about the analysis approach:

1. Spatial persistence

a. I struggled with the various ways that spatial persistence could be interpreted. If patterns are the same between sampling points, then that seems straightforward - the patterns are the same, there hasn't been a change. But if the patterns are different, how do you know if that was because the sampling events were not representative of the first time period or because there was a change in the spatial patterns of inputs (or sinks). If you expect differences in responsiveness to point vs diffuse inputs of nutrients, then it would make sense that spatial patterns would change (assume that point sources and diffuse inputs are all reduced at the same rate [this is unrealistic, but a useful thought exercise], areas most affected by point sources might experience a greater decrease in concentration compared to those most affected by diffuse sources). You are trying to use one analysis to assess both spatial and temporal patterns, but I don’t think you can use this analysis to say whether or not snapshot surveys are representative of temporal trends (your question 2).

b. A second question about the use of these stems from the statement in the text that: “if solutes have low spatial persistence (perhaps due to asynchronous variation) then analyses of patch size and nutrient contribution made for a single moment in time are not reliable indicators of long-term nutrient dynamics.” If low spatial persistence means that you cannot assess things like patch size, were regions/analytes with low persistence excluded from patch size and leverage analyses? Or, would it be interesting to see how patch size and leverage change for those between the two sampling times?

2. Use of ecoregions

a. These ecoregions are really large and encompass a wide range of ecosystems, climate, nutrient management strategies, nutrient management challenges. Some more focused discussion about how you expected patterns to vary among ecoregions based on their hydrology, climate, vegetation, or land use would help justify the use of ecoregions and interpret differences among ecoregions. As is, there are interesting differences among regions, but the reasons for these differences are not explore.

b. I was very confused about the estimates of leverage using the ecoregions. These are not watersheds, so it doesn’t make sense how or why you would calculate leverage for all the catchments within a region relative to the largest catchment in the region. For the temperate eastern forests, the ecoregion “outlet” would be the Mississippi River, but many (most?) of the “subcatchments” are not within the Mississippi River basin, and many of the subcatchments of the Mississippi River basin are not within that ecoregion. It seems to me that using watershed boundaries rather than ecoregions would make much more sense for these analyses which are designed for making comparisons within watersheds.

3. Mass balance approach

a. This really relies on specific discharge being consistent within a watershed or region, which is not the case (as your Figure S4 illustrates well, especially for regions 5,6,7 and 8 in which specific discharge varies by what looks like an order of magnitude).

Specific line comments:

Line 32: how are you defining nutrient resilience? I'm not sure how you reach this conclusion from the analyses

Line 53: Spatiotemporal patterns of what?

Lines 54-57: Can you say more about these trajectories - what are they? Is the point here to refute the legacy hypoth? Or just say that there are more patterns?

Lines 57-59: unclear here exactly what is and isn't known, what the gap is, besides a need for expanded analysis. What will the expanded analysis tell us?

Line 64: unclear what the correlations are between - the locations at diff times?

Line 69: Can you define leverage here?

Lines 77-79: I don’t think you can really address Q2 since you aren't comparing the snapshots with more temporally resolved data

Line 80 (Q4): ok - but you focus your results on the concentration results, not the leverage results. If the leverage results are what are most important, then it might make more sense to put those in the paper and put the concentration results in the supplemental

Lines 113-117: how would other results change if you restricted the 2008 study to include the same range of catchment sizes?

Line 129: boxplot is a visualization tool, not a hypothesis test, but more importantly, unclear what is being compared to what in these boxplots and t-tests

Line 131: is there evidence that nutrient management would vary across these regions?

Do you have specific hypotheses about how patterns of nutrient persistence might vary across the regions? I understand that they are different, but how do you expect those differences to manifest in the response variables that you are focusing on? I'd be concerned that the differences are by chance, rather than because of something relevant to the ecoregion or how the ecoregion was designated.

Line 170-171: Does this mean that other analyses only valid for areas with spatial persistence?

Line 181: a table with the different parameters would be really helpful – how they are calculated, inferences

Line 186-187: What was analyzed for these parameters?

Line 197: are these done for all samples/sites? Or just where there was spatial persistence? Was it done for a single sampling point (early/later?) or all data together?

Line 226: how are you calculating leverage if not all the subwatersheds are within the largest watershed within each region?

Lines 227-229: you found similar specific discharge between dates, but not within ecoregions, which was the assumption that you were testing.

Lines 241-243: these interpretations are based on assumption of consistent within-region specific discharge. How might violations of that assumption (documented in S4, but maybe worth pulling the data that are associated with Q data and seeing how sensitive your results are to this assumption?) affect your interpretations?

Line 252: how were these models selected? What are the important differences among them?

Line 268: what data were missing? Wasn't this dataset based on the water quality data? How many data points were removed for missing values?

Line 285: really over the two 5-year study periods, since you only compared changes over 5 years.

Lines 285-289: reminder needed here - are these based on the paired sites only?

Line 315-316: curious about how you interpret the differences in patch-size between continental and regional analyses - patch is scale-dependent (this isn't really surprising), but what does it mean for nutrient management?

Lines 335-336: shouldn't this be stated, “most catchments had low leverage”?

Line 341: dilution doesn't change the mass balance. I suspect that some of these extreme outliers are due to violations of the assumption that specific discharge is the same within ecoregions.

Line 403: has MAAT been defined? If so, it's been a while and worth spelling out here.

Line 426: how do we know persistence is due to diffuse sources and not point sources (or both)? If concentrations are increasing or staying the same, that might indicate no progress on either front.

Line 431: drivers of nutrient concentrations

Line 435: this seems like a distinct point from the first one, but I'm not sure I understand it.

Lines 445: Or....5 years is not a very long time to measure the impacts of nutrient management. I would be really interested to see an analysis that focuses in on the outliers here. I know the persistence analysis is for the whole dataset (or by region), but could you pull out those sites that changed a lot in rank (or even just look at change in rank as a response variable?) to understand why nutrients persist in some places and not in others. The threshold for persistence seems liberal in my view - at the threshold there is still a lot of change! At any rate, I'm confused about the interpretation of the persistence story, if it's the same, then there hasn't been a change, and you can do all the other analyses, if not then you can say that these are real spatial patterns and do the land use analysis. But I don't think you can use this (esp with a 5 year period) to say that we're not making progress with nutrient management. It's not a long time period and the analysis approach can be interpreted in importantly different ways.

Lines 455-456: also to me means that ALL sources of nutrients have persisted, and that there isn't a difference in the persistence of point and diffuse sources over this time period

Lines 460-462: BUT - this also only captures spatial patterns of nutrient inputs during baseflow. Seasonal high flows or storm events may activate different sources that may or may not be spatially correlated with baseflow sources and may nore may not be spatially persistent.

Lines 479: but differences in patch size between regional and continental scale analyses suggests that patch size varies with scale of analysis. If you had access to data for smaller catchments, you might find yet another answer.

Line 511: Can you define scale of headwaters again here?

Lines 517 onward: Can you link these implications specifically to the model results?

Lines 532-533: spatial patterns were persistent, but the increases in concentrations is relevant too!

Lines 534: can you highlight some of these? Is this new? Comparisons with other analyses?

Fig 2. Is the persistence value for NO3 in region 13 missing? X-axis label missing for panel a

Fig 4a. these are really hard to see because the axes are such large ranges and most of the values are small

Fig 4b. The removal/production don't seem quite right here since you are assuming constant specific discharge. I see the value of making the simplifying assumption, but it does challenge this interpretation

Fig 6 scatterplots: maybe because of the transparency - the wetland points look like the urban color, and other shades are also inaccurate between plot and legend.

how did the year of sampling play out in these models? Would be really neat if you could see how the models vary from year to year

Reviewer #2: Line 86. Is a five year period really adequate to determine temporal trends. Please justify.

Line 398-399. SUrprised ag did not have greater impact on nitrate.

Line 436-438. WHere is the analysis on what the evolution of ag practices was during your time frame? Many areas would have had very little change during your study period. To call out the potential role of ag management changes during this period need to have more information on this.

Line 526-528. THere is little on fertilizer use especially distribution of use and rates. Might be worth calling out there is a need in this area.

Reviewer #3: Some of the supplementary figures embedded in the MS are in EPS format which was inaccessible from my device. Make sure to save it in correct format and check its accessibility. Specifically, check the format of the figures S1 through S10.

6. PLOS authors have the option to publish the peer review history of their article (what does this mean?). If published, this will include your full peer review and any attached files.

Reviewer #1: No

Reviewer #2: No

Reviewer #3: No

---

## [Author Response · Author response to Decision Letter 0]

6 Aug 2021

Dear Reviewers,

Thank you for your careful and thorough reviews. We have addressed each point in the attached Response to Reviewers document.

Thank you,

Rebecca Frei on behalf of all coauthors

---

## [Decision Letter · Decision Letter 1]

28 Sep 2021

PONE-D-21-01607R1Limited progress in nutrient pollution in the U.S. caused by spatially persistent nutrient sourcesPLOS ONE

Dear Dr. Frei,

Thank you for submitting your manuscript to PLOS ONE. After careful consideration, we feel that it has merit but does not fully meet PLOS ONE’s publication criteria as it currently stands. Therefore, we invite you to submit a revised version of the manuscript that addresses the points raised during the review process.

ACADEMIC EDITOR COMMENTS:

Dear Dr. Frei,

You have done an outstanding job addressing most of the reviewer's comments, except for one comment on providing a solid justification for using ecoregions rather than watersheds. I agree with the reviewer's comment that using watersheds rather than ecoregions may have allowed you to compare and make interpretations across regions. However, at this point in the review, I am asking that you include the limitations of using ecoregions and make a point that future studies should use watersheds for the reasons suggested by the reviewer and others.

I hope you can address this comment quickly so that the manuscript can move forward to the next step.

Best wishes.

Gurpal

We look forward to receiving your revised manuscript.

Kind regards,

Gurpal S. Toor, Ph.D.

Academic Editor

PLOS ONE

Journal Requirements:

Reviewers' comments:

Reviewer's Responses to Questions

**Comments to the Author**

1. If the authors have adequately addressed your comments raised in a previous round of review and you feel that this manuscript is now acceptable for publication, you may indicate that here to bypass the “Comments to the Author” section, enter your conflict of interest statement in the “Confidential to Editor” section, and submit your "Accept" recommendation.

Reviewer #1: (No Response)

Reviewer #3: All comments have been addressed

2. Is the manuscript technically sound, and do the data support the conclusions?

Reviewer #1: Partly

Reviewer #3: Yes

3. Has the statistical analysis been performed appropriately and rigorously? 

Reviewer #1: Yes

Reviewer #3: Yes

4. Have the authors made all data underlying the findings in their manuscript fully available?

Reviewer #1: (No Response)

Reviewer #3: Yes

5. Is the manuscript presented in an intelligible fashion and written in standard English?

Reviewer #1: Yes

Reviewer #3: Yes

6. Review Comments to the Author

Reviewer #1: The authors have done a thorough revision, and the resulting manuscript is much stronger and clearer. I was also excited to see the inclusion of new data and the longer time period of their analysis. They did a great job revising the introduction and discussion to better match their analyses, and the methods now provide better detail about the many metrics that they use in the paper. The revisions address nearly all of my concerns.

My one remaining concern is the use of ecoregions as the unit of analysis for the leverage analysis. I would suggest that your rationale for using them include the good reasons that you can't use watersheds - but it doesn't mean that ecoregions are an appropriate substitute. The analysis doesn't make physical sense if the watersheds being compared aren't hydrologically connected. I don't see how you can make any valid inferences from this analysis. I'm also confused by your third point - many of the ecoregions you use are very large and don't have similar land use or climate (e.g., Maine and Florida). While you acknowledge this in the methods, the results and discussion interpret the data as if these were nested watersheds – referring to smaller watersheds as “subwatersheds” and making inferences about nutrient retention between non-nested watersheds. I don’t think the leverage analysis is critical to the story here, and I would suggest cutting it. Otherwise, I think the paper is really nice.

For the trends, I’m curious whether the authors think these are due to variation in nutrient inputs and retention alone, or how much could be caused by interannual variability in hydrology. This could be a small point for the discussion.

Reviewer #3: (No Response)

7. PLOS authors have the option to publish the peer review history of their article (what does this mean?). If published, this will include your full peer review and any attached files.

Reviewer #1: No

Reviewer #3: No

---

## [Author Response · Author response to Decision Letter 1]

4 Oct 2021

Please see Response to Reviewers document.

---

## [Editor Report · Decision Letter 2]

11 Oct 2021

Limited progress in nutrient pollution in the U.S. caused by spatially persistent nutrient sources

PONE-D-21-01607R2

Dear Dr. Frei,

We’re pleased to inform you that your manuscript has been judged scientifically suitable for publication and will be formally accepted for publication once it meets all outstanding technical requirements.

Kind regards,

Gurpal S. Toor, Ph.D.

Academic Editor

PLOS ONE

Additional Editor Comments (optional):

Congratulations!
---

## [Editor Report · Acceptance letter]

16 Nov 2021

PONE-D-21-01607R2 

Limited progress in nutrient pollution in the U.S. caused by spatially persistent nutrient sources 

Dear Dr. Frei:

I'm pleased to inform you that your manuscript has been deemed suitable for publication in PLOS ONE. Congratulations! Your manuscript is now with our production department. 

Kind regards, 

on behalf of

Dr. Gurpal S. Toor 

Academic Editor

PLOS ONE